# Cryptic genetic variation enhances primate L1 retrotransposon survival by enlarging the functional coiled coil sequence space of ORF1p

Anthony V. Furano[1]*, Charlie E. Jones[1], Vipul Periwal[2], Kathryn E. Callahan[1¤a], Jean-Claude Walser[1¤b], Pamela R. Cook[1¤c]

1 Laboratory of Cellular and Molecular Biology, NIDDK, National Institutes of Health, Bethesda, Maryland, United States of America, 2 Laboratory of Biological Modeling, NIDDK, National Institutes of Health, Bethesda, Maryland, United States of America

☉ These authors contributed equally to this work.
¤a Current address: San Rafael, California, United States of America
¤b Current address: ETH Zurich, Genetic Diversity Centre, Zurich, Switzerland
¤c Current address: Department of Biomedical Sciences, Mercer University School of Medicine, Macon, Georgia, United States of America
* anthonyf@helix.nih.gov

**Data Availability Statement:** All relevant data are within the manuscript and its Supporting Information files.

## Abstract

Accounting for continual evolution of deleterious L1 retrotransposon families, which can contain hundreds to thousands of members remains a major issue in mammalian biology. L1 activity generated upwards of 40% of some mammalian genomes, including humans where they remain active, causing genetic defects and rearrangements. L1 encodes a coiled coil-containing protein that is essential for retrotransposition, and the emergence of novel primate L1 families has been correlated with episodes of extensive amino acid substitutions in the coiled coil. These results were interpreted as an adaptive response to maintain L1 activity, however its mechanism remained unknown. Although an adventitious mutation can inactivate coiled coil function, its effect could be buffered by epistatic interactions within the coiled coil, made more likely if the family contains a diverse set of coiled coil sequences—collectively referred to as the coiled coil sequence space. Amino acid substitutions that do not affect coiled coil function (*i.e.*, its phenotype) could be "hidden" from (not subject to) purifying selection. The accumulation of such substitutions, often referred to as cryptic genetic variation, has been documented in various proteins. Here we report that this phenomenon was in effect during the latest episode of primate coiled coil evolution, which occurred 30–10 MYA during the emergence of primate L1Pa7–L1Pa3 families. First, we experimentally demonstrated that while coiled coil function (measured by retrotransposition) can be eliminated by single epistatic mutations, it nonetheless can also withstand extensive amino acid substitutions. Second, principal component and cluster analysis showed that the coiled coil sequence space of each of the L1Pa7-3 families was notably increased by the presence of distinct, coexisting coiled coil sequences. Thus, sampling related networks of functional sequences rather than traversing discrete adaptive states characterized the persistence L1 activity during this evolutionary event.

**Funding:** This work was funded by the Intramural Research Program of the National Institute of Diabetes, Digestive and Kidney Diseases, of the National Institutes of Health. The funders had no role in study design, data collection and interpretation, or the decision to submit the work for publication.

**Competing interests:** The authors have declared that no competing interests exist.

## Author summary

Mammalian L1 retrotransposons replicate by copying their RNA into genomic DNA. Despite being deleterious, a single lineage of successive L1 families emerged in most mammalian genomes, each amplifying before undergoing extinction and replacement by another active family. During ~80 million years of primate evolution this process generated ~40% of the human genome where L1 remains active. Thus, accounting for the persistence of L1 is a major issue. Emergent L1 families are often associated with episodes of extensive amino substitutions in the L1 encoded ORF1p protein, which is required for L1 replication. These bore the signature of positive selection (more amino acid substitutions than expected by chance), which often indicates an adaptive change, implying an "arms race" between L1 and its host. Determing the contestants in this arms race would reveal a major aspect of L1/host interaction. But our findings now suggest an alternative evolutionary model. Most of the substitutions did not affect ORF1p function, and being "hidden" from selection their accumulation could increase sequence diversity (sequence space) of functional ORF1p, which we demonstrated by principal component analysis. The availability of multiple functional ORF1p sequences could buffer ORF1p activity from random inactivating mutations, an evolutionary strategy that could ensure L1 survival.

## Introduction

L1 (LINE-1) retrotransposons are genomic parasites of ancient lineage [1] and have remained active in most mammals over the last 80–120 Myr. L1 replicates by copying its RNA transcripts and those of other genes into genomic DNA and by now L1 activity has generated $\sim 40\%$ of the human genome [2–10]. Individual L1 families can contain thousands of copies, although most are 5'-truncated and otherwise defective on insertion. Despite their potential for seriously deleterious effects [11–13], being subject to strong negative selection [14–16] and susceptible to host repressive mechanisms [reviewed in 17], L1 is now the dominant active retrotransposon in many mammals, including humans, represented by the active L1Pa1 (L1Hs) family. The human genome retains a trove of L1 fossils, the relics of a series of extinct L1 families that were at one time ascendant in the primate lineage, which provide a historical record of the evolutionary antecedents of L1Pa1.

Accounting for the persistence of L1 remains a major issue for mammalian biology. Foundational, pre-whole genome studies in rodents and primates showed that evolutionary change of the 5' UTR regulatory region or the amino (N)-terminal half of ORF1p [18–20] and [reviewed in 21] was often associated with the emergence of novel L1 families. These studies and subsequent bioinformatic analyses showed that competition for limiting host factors or bypassing host repression likely drives the recruitment of novel 5' UTRs [7, 22–24], also see [17]

In contrast, the basis of variability of the N-terminal half of ORF1p is unknown. This region is dominated by an evolutionary labile coiled coil domain, a predicted motif in all vertebrate L1 families [1], which contrasts with a highly conserved carboxy (C)-terminal half that mediates properties associated with retrotransposition: high affinity nucleic acid binding, nucleic acid chaperone activity, and rapid formation of stable nucleoprotein complexes(Fig 1A, [25, 26, and references therein]). Coiled coil variation can occur by different means: In mice and rats by addition or deletion of heptad repeats [18–20, 23]; in primates, by episodes of extensive amino acid substitutions with a signature of positive selection (more amino acid substitutions

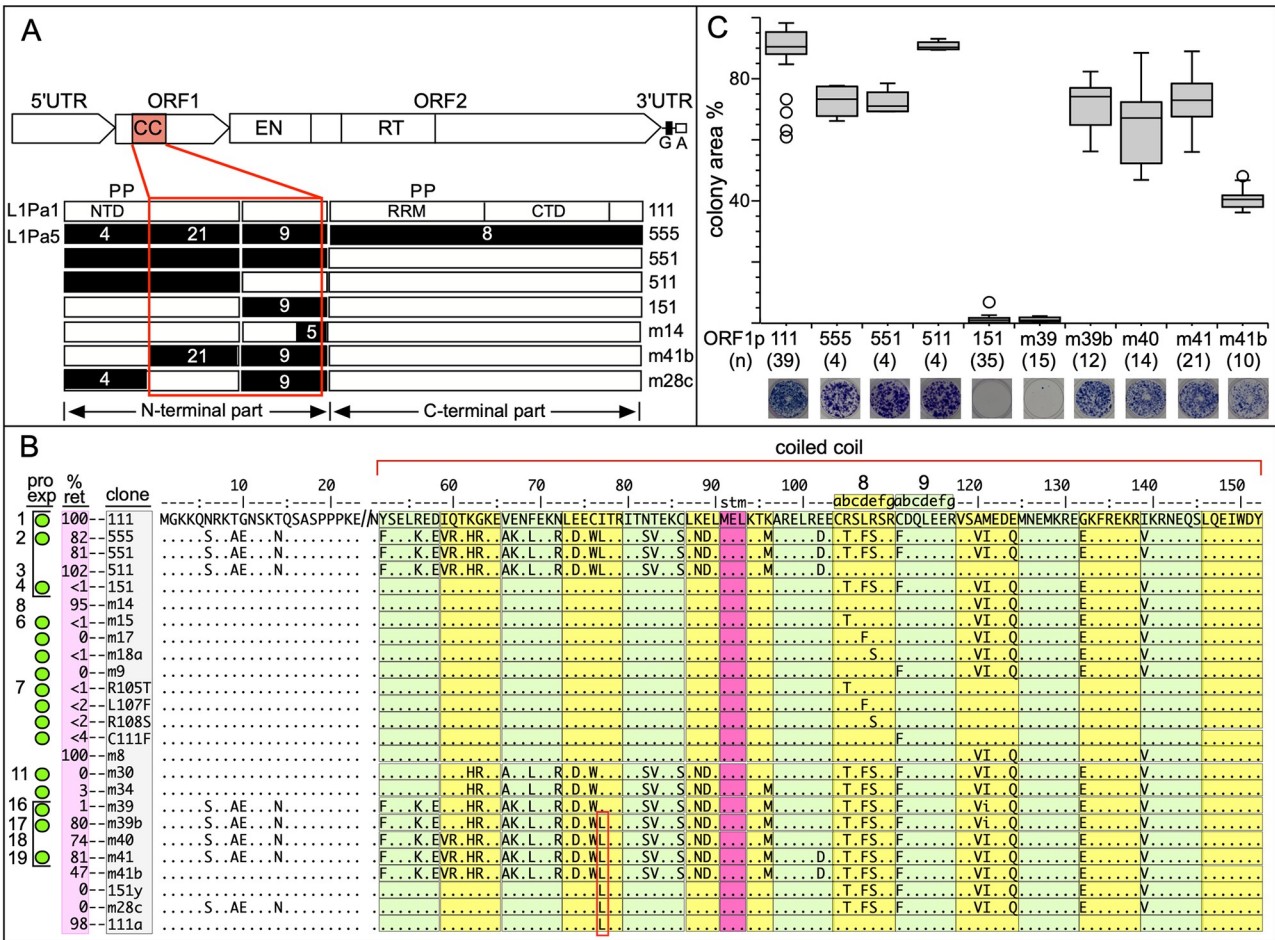

**Fig 1. Retrotransposition of ORF1p variants. A**—Generic L1 retrotransposon [1, 43]: 5' UTR (untranslated regulatory region); ORF1 (open reading frame1) with coiled coil domain (CC), which mediates trimerization of ORF1p necessary for high affinity nucleic acid binding and chaperone activity [40, 44–46]; ORF2, encodes the L1 replicase with endonuclease (EN) and reverse transcriptase (RT) domains; 3' UTR with a conserved G-tetraplex forming domain (G) [47, 48] and an A-rich tail (A). P, location of conserved phosphorylation sites in mammalian ORF1p required for retrotransposition [1, 49]. NTD, RRM, CTD, N-terminal domain, RNA recognition motif, C-terminal domain, respectively. Also depicted are schematics of 8 ORF1p sequences 7 of which are mosaic structures consisting of the indicated regions of a modern active L1Pa1 (L1Hs) ORF1p (white, 111) and a resuscitated ancestral L1Pa5 ORF1p (black, 555) [26]. The names of the variants are given on the right and the white numbers indicate their amino acid differences. **B**—Alignment of the N-terminal 23 amino acids and the entire 14 heptad coiled domain (alternating green and yellow boxes). Note, the heptads are numbered from 1–14 (only 8 & 9 are indicated) so as to be congruent with amino-carboxy orientation of the protein. L1Pa1 ORF1p is the reference sequence and dots and letters indicate respectively identities and differences between it and the other variants. Four columns are listed on the left: Numbers, corresponding to the subset of the variants shown in S3 Fig that were mapped on the coiled coil sequence space shown in Fig 2; pro exp—green dots, variants tested for protein expression in HeLa cells. Both active and inactive protein were expressed (S1 Fig); % ret—% retrotransposition activity in HeLa cells; clone—variant names. The a-g heptad amino acid positions are shown for heptads 8 and 9, stm indicates the stammer in heptad 6. **C**—Box plots of retrotransposition assays of selected variants (bracketed in panel **B**) and representative stained G418 resistant foci and the numbers (n) of independent transfections (biological replicates) are indicated. Retrotransposition results of the other variants are shown in S2 Fig.

than expected by chance) [7, 27]. This last occurred 30–10 MYA during evolution of the primate L1Pa7-L1Pa3 families [7, 27]. In both mice and human, instances of coiled coil variation have been interpreted as an adaptive response to genetic changes extrinsic to ORF1p [23, 27].

However, here we present results that indicate an alternative model of coiled coil evolution at least in primates. We found that retrotransposition activity can tolerate extensive coiled coil amino acid changes. Such phenotypic indifference (termed genetic robustness) could be permissive to the accumulation of cryptic genetic changes, so called because they would be not

subject to ("hidden" from) selection [28–34]. This condition could have increased coiled coil diversity (*i.e.*, expand the coiled coil sequence space). Principal component analysis (PCA) along with cluster and phylogenetic analyses showed that this did occur and revealed that the coiled coil domains of each of the L1Pa7 through L1Pa3 families consist of three or more related, co-existing but distinct coiled coil sequences, some of which persisted in descendant L1 families. One of the three L1Pa3 coiled coil sequences survived in the active L1Pa1 family of modern humans and its immediate precursor L1Pa2 family, which remains active in chimpanzee and bonobo [35, 36].

Thus, evolution of the L1Pa7 –L1Pa3 coiled coils did not occur solely by stepwise progression through family-specific adaptive states as we and others originally proposed [1, 7, 27], but by exploration of a variety of functional coiled coil sequences. This model of coiled coil evolution has dramatically different implications for host / L1 interaction than an adaptive model. Furthermore, an expanded functional sequence space of the coiled coil would favor L1 survival, by buffering coiled coil function [37, 38] from adventitious, retrotransposition-destroying coiled coil substitutions [this study, and 39, 40]. Such profound epistasis is not unique to L1 ORF1p but is a feature of coiled coil genetics [41, 42]. What remains unanswered is what initiates an intense episode of coiled coil change during L1 evolution.

## Results

### ORF1p activity can tolerate extensive coiled coil amino acid substitutions but is sensitive to single epistatic substitutions

In order to address the functional consequences of the last episode of intensive coiled coil amino acid change that occurred 30–10 MYA we had resuscitated the ORF1 sequence of the now extinct L1Pa5 family (Materials and methods) and [26], which had attained peak retrotransposition activity ~25 MYA. We refer to its encoded ORF1 protein as 555p, and as Fig 1A shows, 30 of the 42 amino acids that distinguish 555p and its L1Pa1 encoded modern counterpart, 111p, are located in the coiled coil. Fig 1B and 1C show that substituting 555p for modern 111p only modestly affected retrotransposition of the modern L1Pa1 element. Furthermore, mosaic ORF1 proteins consisting of an ancestral amino terminal half, 551p, or a coiled coil, which contains 21 of the 30 ancestral residues (511p) were similarly as active as the modern protein in supporting retrotransposition. And finally, an ORF1p in which all of the modern coiled coil amino acids were replaced by their ancestral counterparts (30 residues, variant 41b), retained about half of its retrotransposition activity.

Although 151p is inactive for retrotransposition, it is expressed in HeLa cells at levels comparable to 111p (S1 Fig) and [26]. Furthermore, 151p is essentially indistinguishable from 111p and 555p in trimer formation and several oligonucleotide-based assays: affinity for nucleic acids, nucleic acid chaperone activity, and stabilization of mismatched oligonucleotide duplexes [26]. However, single molecule binding to long nucleic acids revealed the defect in 151p –while 111p and 555p trimers rapidly convert to stably bound oligomers after binding single stranded DNA, 151p trimers are unable to do so. We had suggested that the 151p coiled coil prevents its trimer from assuming a conformation that mediates its oligomerization on nucleic acids [26].

Thus, one or more of the 9 ancestral coiled coil amino acids in 151p is negatively epistatic in the presence of modern residues in this protein. Stepwise replacement by their modern counterparts revealed that any one of the 4 ancestral residues in heptads 8 or 9 (positions 105, 107, 108, or 111) either in the presence of the 5 downstream ancestral residues in heptads 10, 12 and 13 (variants m15, m17, m18a, m9) or on their own (variants R105T, L107F, R108S, or C111F) inactivates ORF1p. In every case the single substitution essentially abolishes

retrotransposition, recapitulating the phenotype of the 151p protein. The 5 ancestral residues in heptads 11, 12, and 13 do not affect ORF1p activity (variant m14, Fig 1). Representative retrotransposition assays supporting these conclusions are shown in Fig 1B and S2 Fig. The green dots (column 2, pro exp, Fig 1B) indicate the ORF1p variants that we tested for expression in HeLa cells. S1 Fig shows that a lack of expression does not account for the retrotransposition null phenotype.

We then proceeded to restore retrotransposition competence to 151p by stepwise replacement of its modern residues to their ancestral counterparts. Fig 1B shows that restitution of the partially ancestral 151p coiled coil to an active phenotype, variant (m39b), required 18 of 21 ancestral residues in heptads 1–8, and full activity was attained only upon insertion of the 18$^{th}$ ancestral amino acid (I77L, red rectangle, Fig 1B). Until then, the protein was essentially inactive for retrotransposition. Furthermore, L77 on its own (variant 151y) did not affect the activity of 151p; or a version of 151p that contained the 4 ancestral residues in the amino terminal domain (NTD, variant m28c); or the fully modern protein, variant 111a. Thus, the rescue of ORF1p activity by I77L depends on the coiled coil context afforded by most of the other ancestral residues, an instance of epistasis just as profound as the inactivating single substitutions in heptads 8 and 9. S3 Fig shows the effect on retrotransposition of all of the coiled coil substitutions that we examined.

Fig 1 shows that while some of the 30 substitutions that differentiate modern and ancestral coiled coils could be considered biochemically "conservative" (*i.e.*, amino acids of similar charge, polarity, hydrophobicity, etc.) others are not. Thus, in terms of the coiled coil-endowed property that renders ORF1p active in retrotransposition, the coiled coil is "genetically robust" *i.e.*, it can be phenotypically indifferent to a considerable degree of genetic change. Genetic robustness would be permissive to the accumulation of cryptic genetic changes—termed "evolvability" [29], which are only manifested by the effect of epistatic substitutions as illustrated by the above examples. Theoretical and experimental studies indicate that sampling functional sequences that had been generated by a combination of randomly acquired cryptic and epistatic mutations provide an alternative model of evolutionary change to the stepwise traverse of different adaptive genetic states [28–33]. Therefore, we determined the coiled coil sequence space encompassed by all the members of the L1Pa7—L1Pa1 families as outlined in the next section and described in the Materials and Methods.

## The coiled coil sequence space of the L1Pa7—L1Pa1 families

We determined coiled coil sequence space by subjecting coiled coils alignments to two analytical techniques that provide graphical views of their sequence relatedness (see Materials and methods). The first, which generated a three-dimensional view of the coiled coil sequence space, was principal component analysis (PCA) of a single global alignment of the coiled coils. A unique 20-bit binary vector (one-hot encoding) was assigned to each amino acid, and each family was assigned a different color. Fig 2A shows that the entire ensemble of coiled coils traverses a contiguous sequence space, each of which, with the exception of L1Pa2 and L1Pa1, is somewhat diffuse and elongated. Mapping the coiled coil variants on this space (numbered as in Fig 1B and S3 Fig, large circles indicating active variants) graphically illustrates the conclusions that we made from Fig 1: Genetic robustness—variant 3 maps well out of the sequence space but is fully active. Strong epistasis—the paired active and inactive variants (1 & 7; 16 & 17) share the same sequence space.

We also used metric multidimensional scaling (MMDS), as implemented in the Bios2mds R package [50], which we applied to the difference matrices of aligned coiled coils of each family (see Materials and methods). This method provides a two-dimensional graphical view of

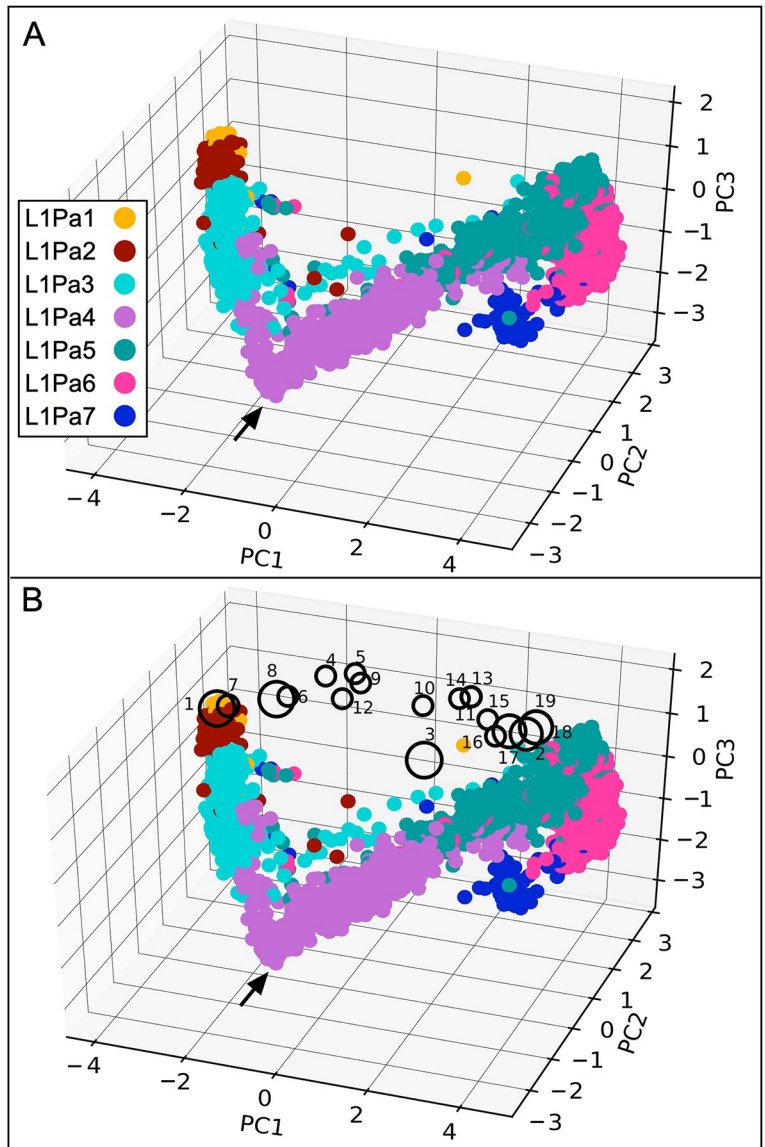

**Fig 2. Principal component analysis of the ORF1p coiled coil. A**—The color code used for each family corresponds to one of the major clusters shown in Fig 3. **B**—Large and small circles correspond respectively to active and inactive ORF1p variants mapped on the sequence space of the coiled coil of L1pa7-L1Pa1 and are numbered per Fig 1 and S3 Fig. Active variants (large circles) exhibited ~80–100% of L1Pa1 activity and inactive ones (small circles) <5% of L1Pa1 activity.

sequence diversity within each family. Fig 3 (cluster analysis, left side) shows that except for L1Pa2 and L1Pa1, K-means based clustering resolved the sequence space of the coiled coils of each L1 family into three or more clusters of varying compactness. The clusters are designated cLn.n, indicating the cluster and family numbers respectively; *e.g.*, cL1.7 is cluster 1 of L1Pa7. In contrast to the coiled coils, there was little substructure in the sequence space of the L1Pa7 – L1Pa3 carboxy terminal half of ORF1p (S4 Fig). We retrieved the sequences comprising the individual coiled coil clusters and found that the 50% consensus sequences of clusters cL3.5 and cL1.1 correspond respectively to the coiled coils of the 555p and 111p ORF1p sequences (Fig 1).

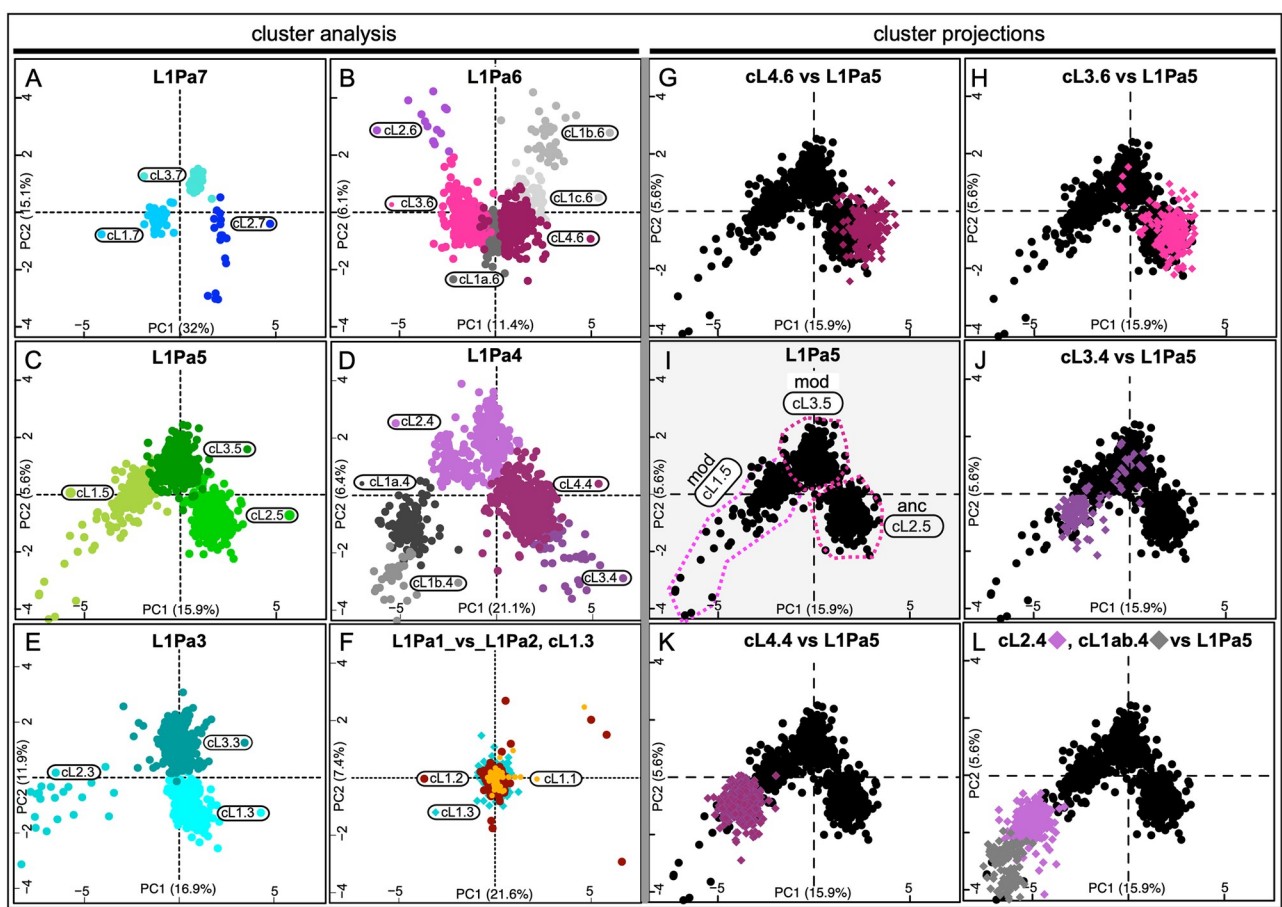

**Fig 3. Cluster analysis of coiled coil sequence space.** Panels A-F. Coiled coil clusters identified in L1Pa7 –L1Pa1 by the bios2mds R package [50] as described in the Materials and Methods. Clusters are designated as follows: cLn.n, the cluster number followed by the family number—cL3.7 is cluster 3 of the L1Pa7 family. Panel F shows the projection of L1Pa1 (cL1.1) on the sequence space of L1Pa2 (cL1.2) and cL1.3, using the **mmds.project** function of the Bios2mds package as described in the Materials & Methods. Panels G-L show projection of L1Pa6 or L1Pa4 clusters on the coiled coil sequence space of L1Pa5. Panel I shows 3 coiled coil clusters for L1Pa5: $cL1.5_{mod}$, $cL3.5_{mod}$, and $cL2.5_{anc}$, which belong to the modern and ancestral versions of L1Pa5 (see text). The 50% consensus sequence of the cL3.5 cluster corresponds to the 555 ORF1p sequence, marked with an asterisk, *, on Figs 4 and 5.

We used two methods to examine the relationship between coiled coils of different L1 families: (1) the **mmds.project** function of Bios2mds; (2) phylogenetic analysis using maximum likelihood. The first method projects the coiled coil sequences of a given cluster onto the sequence space of another cluster or family. Fig 3G–3L show examples of this analysis wherein we projected the 2 largest L1Pa6 clusters or all of the L1Pa4 clusters onto the L1pa5 family (Fig 3I). The L1Pa6 clusters showed varying degrees of overlap with sequence space of the ancestral (anc) cL2.5 cluster, and the L1Pa4 clusters exhibited different degrees of overlap with the sequence space of the modern (mod) L1Pa5 clusters., cL1.5 and cL3.5. The mod version of L1Pa5 ORF1p (Fig 3I) and its descendant L1pa4–L1Pa1 ORF1p sequences lack a tripeptide located 2 residues upstream of the coiled coil that is present in anc L1Pa5, L1Pa6 and L1Pa7 (see S1 Data/orf1_FL/l1pa5). However, the anc and mod L1Pa5 C-terminal half of ORF1p map to the same sequence space, S4 Fig. Whereas multiple coiled coil clusters populate the L1Pa7 – L1Pa3 families, only one cluster populates the L1Pa2 and L1Pa1 families. The coiled coils of these families and cL1.3 map to the same sequence space (Fig 3F).

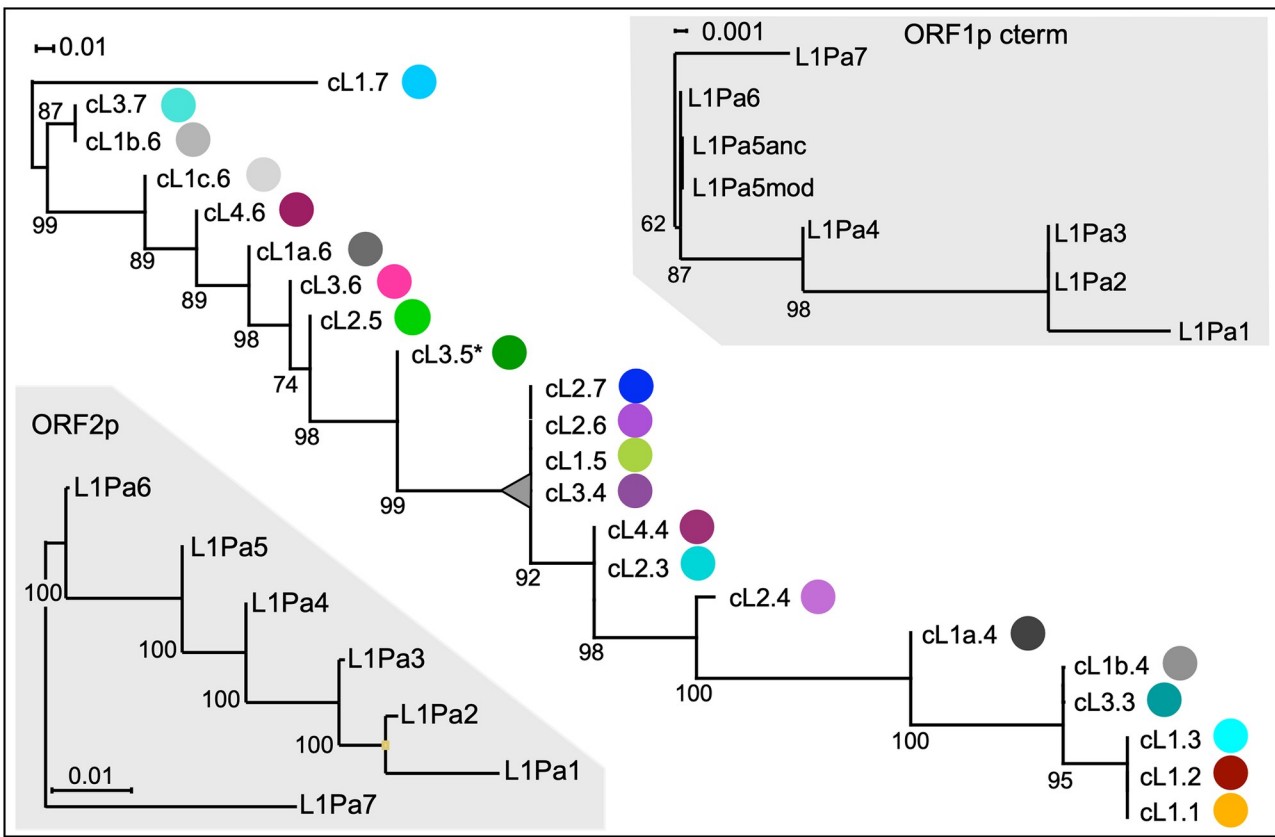

**Fig 4. Coiled coil phylogeny.** Maximum likelihood trees of the coiled coil clusters and C-termini were built on their 50% consensus sequences with the amino acids encoded by CG-affected codons treated as missing data. Note the 10-fold lower scale of the branch lengths for the C-terminus tree. The colored circles at the tips of the coiled coil cluster tree correspond to the cluster colors in Fig 3, panel A-F. The numbers at each node give its frequency as % of 1000 bootstrap replicates. The ORF2 tree was generated from amino acid consensus sequences of the human version of our previously described collection of L1Pa2 –L1Pa7 human/chimpanzee orthologues [51] and the currently active human L1Pa1 family (in particular, Ta1-d 5), represented here by the L1.3 element [52]. This tree is consistent with a previously described tree built from the 3' 2 kb of nucleic acid sequence which includes the 3'UTR but mostly ORF2 sequence (Figure 4A in [7]).

Fig 4 shows that phylogenetic analysis of the 50% consensus sequences of the clusters (see Materials & methods) corroborated and extended the foregoing results. As expected, the coiled coil consensus sequences of cL1.3, L1Pa2 and L1Pa1 map to the same node. But unexpectedly, some ancestral coiled coil sequences persisted over several generations of L1 families–*i.e.*, some coiled coil clusters recovered from the L1Pa7 –L1Pa3 families shared several (highly supported) nodes on the maximum likelihood tree. These are bracketed on Fig 5, which shows an alignment of the consensus sequence of each cluster in reference to the most ancestral cluster, cL1.7. Therefore, coiled coil clusters present in the L1Pa7 and L1Pa6 families, were still being propagated in the ORF1p sequences of the L1Pa5 and L1Pa4 families (large grey triangle in Fig 4). Thus, the coiled coil phylogeny differs markedly from that of ORF2p, which shows the typical single lineage of successive mammalian L1 families that emerged, amplified and went extinct[1]. Also, note the 10-fold lower scale for the branch lengths of the ORF1p C-terminus tree. Taken together these results indicate that the ORF1p coiled coil, its C terminal half, and ORF2p are evolving under markedly different constraints (Fig 4).

**Fig 5. Coiled coil amino acid changes.** The 50% consensus sequence of cL1.7, is at the top of the alignment and those of the other clusters given below, arranged according to their position on the phylogenetic tree (Fig 4), with the ones sharing the same node bracketed. The number of sequences in each cluster is given in the right-hand column. Dots indicate amino acid identity, and letters indicate differences, capitalized upon their first appearance. The thin and heavy underlined positions of the cL1a.4 consensus indicate respectively the modern (111) residues that had already arisen in the coiled coil or first appeared here. The pink and blue columns highlight the emergence and replacement of the residues in heptads 8 and 9 that are negatively epistatic in the modern coiled coil (Fig 1). The red box shows the emergence of I77, which is negatively epistatic in the ancestral context. The sequences of 111 with its differences from the active 555 and m39b variants and the inactive m39 variant are at the bottom of the alignment. S5 Fig shows the CG-less (_o) and CG-restored (rt) translation products of the coiled coil clusters consensus sequences. S6 Fig shows a LOGO plot of the CG-restored translation products, and S7–S11 Figs show various alignments of the coiled coil sequences that populate the cL1.3 (L1Pa3), L1Pa2 (cL1.2) and L1Pa1 (cL1.1) clusters (see Discussion).

## Relationship between the effects of experimentally introduced coiled coil substitutions and the record of evolutionary change in the coiled coil clusters

Fig 1 and S3 Fig show that each of four ancestral amino acids (T105, F107, S108, and F111) were negatively epistatic in an otherwise modern coiled coil. Furthermore, as mentioned earlier, step wise restitution of ancestral residues in the presence of this ancestral quartet (151p) had no effect until the 18[th] (I77L) of the 21 ancestral coiled coil amino acids (that separate inactive 151p and fully active 511p) was introduced, and it completely restored activity. The alignment of the coiled coil cluster consensus sequences in Fig 5 shows that the ancestral quartet was present in L1Pa6 and older families (purple vertical bars). They were replaced by their modern counterparts (blue vertical bars) at about the same time and roughly coincident with the emergence of cL1a.4. By this time most of the modern residues were either already present (underlined) or coincident with their appearance in cL1a.4 (heavy underline). Furthermore, the replacement of L77 by I, which is strongly negatively epistatic in an ancestral context (Fig 1, m39 vs m39b), was also approximately coincident with the above substitutions (cf. red and

blue columns in Fig 5). These sequence changes approximately correspond to the sharp inflection in the direction of the sequence space (arrow, Fig 2) that occurred in the evolutionary path to L1Pa2 and L1Pa1. Thus, the experimentally introduced substitutions, which were carried out before, and thus without regard to the evolutionary analysis, in effect recapitulates its end result if not its chronology.

## Discussion

We demonstrated the relative indifference of L1 retrotransposition activity to extensive amino acid substitutions in the ORF1p coiled coil (Fig 1, S3 Fig). Earlier studies showed that such lack of phenotypic effect (genetic robustness) could increase genetic diversity, which could be both revealed by, and buffer the effect of, epistatic mutations [28–34]. We found that such phenomena were associated with the episode of intense coiled coil change during the evolution of the L1Pa7-L1Pa3 families. In particular, the results in Figs 2–5 show that the expanded coiled coil sequence space could benefit L1 survival [37, 38] by buffering ORF1 function from adventitious, retrotransposition-destroying coiled coil substitutions [this study, and 39, 40]. Such events are not just theoretical possibilities based on instances of epistasis exhibited by other coiled coils. Fig 1B shows strong epistatic effects of single amino acid substitutions that either completely inactivated or fully revived retrotransposition. Taken together our results provide a mechanistic explanation for the instance of rampant coiled coil amino acid substitutions that occurred during the evolution of the L1Pa7 –L1Pa3 families.

The coiled coil mediates trimerization of ORF1p monomers, and the trimer is the active form of the protein in retrotransposition. Only trimers exhibit high affinity nucleic acid binding and nucleic acid chaperone activity, and ORF1p that lacks either property cannot support retrotransposition[40, 44–46]. However, trimer formation *per se* is not sufficient. The 151p trimer is essentially indistinguishable from the 111p trimer by both biophysical parameters [53] and nucleic acid binding and chaperone activity using oligonucleotide-based assays [26], but it is inactive for retrotransposition. Therefore, a "generic" trimer is not sufficient; it also has to license the inter-trimer contacts between the C-terminal half of the protein that supports their rapid polymerization on single-stranded nucleic acid [26].

Figs 2 and 3 indicate that each of the five families that emerged and went extinct during evolution of L1Pa7-L1Pa3 propagated related but distinct coiled coils. So, had the different coiled coil sequences amplified concurrently or sequentially during the life of the family? The persistence of coiled coil clusters 2.7, 2.6, 1.5 and 3.4 through 4 generations of L1 families (Figs 4 and 5) suggest they amplified concurrently and that coiled coils maintained their identity in the presence of co-existing coiled coils.

This finding poses several issues: If hybrid trimers (*i.e.*, those composed of monomers from different L1 clusters) can form, they would have to produce replication competent ORF1p in order to be propagated. The possibility of forming hybrid trimers could be addressed by co-expressing differentially tagged coiled coil sequences. On the other hand, hybrid trimer formation would not be an issue if L1 protein synthesis is functionally compartmentalized, which is implied by the concept of *cis* preference, whereby retrotransposition-competent L1 proteins bind to their encoding transcript. This phenomenon has been fairly well established both theoretically and experimentally for transposable elements (and some viruses) including L1 [see 54, and references 13–16 therein]. How cis preference is mediated is unknown, but there is evidence that translation can be sequestered in subcellular compartments or translation factories [e.g., 55].

A second, and puzzling issue is the apparent episodic nature of coiled coil variation (see Introduction). Presumably, when a more fit coiled coil emerges, L1 elements harboring it can

out compete coexisting L1 elements, which at times can mark a change in the path of the coiled coil sequence space (*e.g.*, arrow in Fig 2). Thus, the current lull in coiled coil variation may merely reflect the fact that insufficient time has elapsed for novel coiled coils to be visible above the background of existing coiled coils. The emergence of a novel coiled cluster is exemplified by cluster 1 of L1Pa3 (cLs1.3). This coiled coil and those of the L1Pa2, and L1Pa1 families have identical consensus sequences (Fig 5 and S5 Fig). To determine whether the coiled coils of cLs1.3, L1Pa2 and L1Pa1 are undergoing differentiation we used two methods that are more sensitive than PCA determined by one-hot encoding or MMDS. These are phylogenetic analysis [maximum likelihood 56] and the cluster_fast command of the usearch V11 suite [57]. Setting the id (identity) parameter of the cluster_fast command to 1.00 would recover coiled coil clusters that are minimally divergent from their consensus. Neither method revealed distinct clades within these coiled coils. However, sequence alignments showed that the non-CG encoded F (phenylalanine) at position 134 is hypervariable in about 10% of L1Pa2 family, generating mostly substitutions to valine (V) or serine (S) due to single nucleotide transitions or transversions in the F codons (TTT or TTC). Such variants were relatively rare in the descendant L1Pa1 family, indicating that they were not propagated or retained. These alignments are shown in S7–S11 Figs.

In conclusion, evolution of primate ORF1 coiled coils was subject to countervailing but ultimately complementary genetic events that could result from its genetic robustness; permissiveness to enlarging the coiled coil sequence space, which in turn could preserve coiled coil function by buffering the effects adventitious inactivating epistatic mutations. Although a comprehensive mechanistic explanation for the role of the coiled coil in L1 activity has yet to be achieved [28–34], its evolutionary persistence throughout vertebrates [1] attests to its vital role for L1 activity. The fact that the coiled coil is uniquely hypervariable compared to the remainder of ORF1, and that episodes of intense coiled coil amino acid substitutions in primates have been associated with the emergence of novel L1 families imply that such variability is essential to its survival. Murine rodent coiled coils can also be highly variable, but in this case due to changes in length or number of the repeat units (or both) rather than amino acid substitutions [18–20, 23]. Also, whether different versions of the coil can coexist in the same mouse L1 family or mitigate the effect of negative epistatic substitutions [39] has not been addressed. As coiled coils have been identified in at least 10% of all proteins [41], our findings would be relevant to fields beyond L1 retrotransposons. However, two aspects of L1 biology contributed to our detecting intra-family coiled coil variants: Most L1 families generate hundreds or more copies before being superseded by a successor family, and most of these copies are retained in the genome.

## Materials and methods

### Plasmid DNA

pRTC2—This vector is based on the original retrotransposition reporter [58] but it contains the highly active L1.3 [52] sequence kindly provided by Dr. John Moran on the JCC8 vector [58]. We also made other modifications: The pRTC2 backbone was derived from pCEP4 (Invitrogen Life Technologies). We replaced the NruI—SalI-1989 bp fragment by <u>TCGCGA</u>GA AGTA<u>GGTACC</u>TAAT<u>AAGCTT</u>TCAT<u>GCGGCCGC</u>AGAC<u>CGTACG</u>ATCGAG<u>TCAAGTCGAC</u>, which contains sites for NruI, KpnI, HindIII, NotI, BsiWI and SalI, underlined, left to right. pRTC2 also differs from the original reporter by the following modifications: the CMV promoter that drove sense transcription of the L1 element was replaced with the SV40 early promoter (flanked by KpnI and HindIII) and the anti-sense G418 gene was relocated from its original position within the L1 3' UTR to down-stream of it, and its SV40 early promoter was

replaced with the Rouse sarcoma virus LTR. The annotated sequence of pRTC2 containing the L1.3 L1 sequence (flanked by BsiWI and SalI) is included in S1 Data.

ORF1 mutations were generated by site-directed mutagenesis using QuickChange II (Agilent Technologies) using forward and reverse primer pairs designed with the Agilent Primer Design Program (www.genomics.agilent.com/primerDesignProgram.jsp0) on a vector, MB18-111, described in [26]. This vector contained just the 5'UTR and ORF1 and the mutated ORF1 sequences were isolated as a BsiWI/AgeI fragment, which contained the 5'UTR and encoded all but the C-terminal 10 residues of the 338 amino acid ORF1 sequence. The AgeI site is conserved in 555p and 111p. This fragment was inserted into the corresponding sites of pRTC2_Δ_BsiWI-AgeI.

## ORF1-constructs

The modern version of human ORF 1, ORF1-111, and the ancestral version, ORF1-555, as well as the mosaic modern-ancestral ORF1 constructs 151, 551, and 511 are shown in Fig 1 and described in [26]. This paper describes in detail the resuscitation of the ancestral L1Pa5 ORF1 sequence. Basically, we derived a 60% consensus sequence from an alignment of L1Pa5 ORF1 sequences that we had retrieved from the human genome database. We converted to CG those positions in the consensus that corresponded to the positions in the alignment that contained CG and either TG or CA (usually both). We resolved rare ambiguities by comparing the encoded protein to those encoded by L1Pa6 and L1Pa4 consensus sequences. We restored activity of ORF1-151 by sequential steps of PCR site-directed mutagenesis. Generation of the equivalent of ORF1-551 (*i.e.*, m41 on Fig 1 and S3 Fig) from ORF1-151 required 25 amino acid changes, 21 of which are located within the coiled-coil domain. The remaining four residues are located in what we designated as the N-terminal domain (NTD), Fig 1A. The procedures for constructing the pRTC2 retrotransposition vectors from the intermediate holding vectors described above are given in refs [26, 49].

## pORF1-FLAG

As described in the Supporting Information of [49], the mammalian expression vectors were constructed with pcDNA3.1(+)-puro (from the Don Ganem laboratory, University of California San Francisco). ORF1-Flag amplicons, containing a 5′ BamH1-Kozak sequence and 3′ EcoRI-FLAG sequence, were generated by PCR with a high-fidelity polymerase from WT or mutant ORF1 pRTC2 templates with the forward primer CGCGGATCCGCAATGGGGAAAAAACAGAAC and reverse primer GCCGGAATTCCTACTTGTCGTCGTCGTCCTTATAATCCATTTTGGCATG. The PCR fragment was inserted into pcDNA3.1(+)-puro. Some mutants were made using WT pORF1-FLAG as a template for site-directed mutagenesis. All mutations were verified by DNA sequencing, and plasmid DNA was purified using the endotoxin-free plasmid DNA purification kit, NucleoBond Xtra Midi EF (Macherey-Nagel). These plasmids were used to compare expression of the various ORF1p constructs.

## Retrotransposition assays

The pRTC2 plasmid DNA was amplified in NEB 10-β competent cells (New England Biolabs), and extracted using a midi-prep DNA kit (NucleoBond Xtra Midi EF (MACHERY-NAGEL). HeLa cells (HeLa-JM, kindly provided by John Moran, University of Michigan, Ann Arbor) were plated in 6-well dishes (1x $10^5$ cells in 2 ml) or in 12 well dishes (0.35 X $10^5$ cells in 1 ml) and incubated for 20–24 hours until 60% to 80% confluent. The cells were transfected with 1 μg plasmid DNA and 3 μl Fugene6 Transfection Reagent (Roche) in serum free media for 6-well plates, or 0.5 μg DNA and 1.5 μl Fugene 6 in serum free media for 12-well plates. After

72 hours media was replaced with fresh media containing 400 μg/ml G418 antibiotic (Gibco) and incubations were continued for 8–10 days, replenished as needed with fresh G418-containing media. The cells were washed twice with 1X PBS, fixed with 2% formaldehyde (Mallinkrodt)/0.2% glutaraldehyde (Sigma-Aldrich), washed twice with 1X PBS, and stained with Karyo Max Giemsa Stain (Gibco). The stained cells were sequentially washed with 50% ethanol, 15% ethanol, and then water. The plates were photographed with a Canon EOS Rebel T3i camera body and a Canon Macro LENS EF-S 60 mm 1:2.8 USM lens. The camera was operated with Adobe Photoshop Lightroom 4 software; digital images were quantitated for percent plate coverage by adherent cells, with ImageJ, using the "ColonyArea" plugin [59]. Box plots were generated by KaleidaGraph (v.4.5.2, Synergy Software). At least 4 independent transfections were performed for each assay.

## Western Blot analysis

Expression assays were carried out essentially as described in the Supporting Information for Cook *et al* [49]. HeLa cells, in six-well plates, were transfected with 1μg of pORF1-Flag constructs using 3μL of FuGENE6 (Promega). After 48 h, the cells were washed with PBS, lysed with 50 mM Tris·Cl pH 7.4, 650 mM NaCl, 1 mM EDTA, 1% Triton X-100, cOmplete EDTA-free protease inhibitor mixture (Roche), 100 μM leupeptin, and sonicated in a Bioruptor (Diagneode) and centrifuged at $17,000 \times g$ for 15 min at 4˚. Fifty μg samples of supernatant protein were subjected to denaturing gel electrophoresis, transferred to PVDF membranes using iBlot (Invitrogen), blocked with Superblock T20 buffer (Pierce) and incubated overnight at 4˚ with mouse anti-FLAG M2 monoclonal antibody (Sigma-Aldrich) and rabbit anti-tubulin (Sigma-Aldrich) that had been diluted in Superblock T20 buffer. After rinsing with 1XTBS/0.05% Tween20, the membranes were subjected to three 10 min washes with the same buffer and incubated for 1.5 h at room temperature with mouse and rabbit anti-horse radish peroxidase antibodies in blocking buffer. After four 10 min washes in 1x TBS/0.05% Tween20, followed by three rinses with 1x TBS the membranes were developed with Pierce Pico-west substrate and exposed to film.

## Bioinformatics and sequence analysis

**ORF1 and coiled coil sequences.** L1 sequences corresponding to L1Hs (L1Pa1)–L1Pa7 of sufficient length to include full length ORF1 were identified in repeat masker output files (http://repeatmasker.org/species/hg.html) for hg19 (build 37) and hg18 (build 36.1). The corresponding L1Hs and L1Pa2 sequences were retrieved from hg19 using the bedtools getfasta script (2.26.0, http://quinlanlab.org), and L1Pa3 –L1Pa7 sequences from hg18 (indexed by formatdb with the–o option) using the blastall program, fastacmd. The sequences were aligned using Muscle [60] as implemented either in SeaView [61] or Biowulf, the NIH HPC system. ORF1 sequences were isolated from these alignments and those with large inserts or deletions were discarded. Sequences corresponding to the coiled coil and carboxy terminal half were located by reference to these regions of L1Pa1 (L1Hs) and our resuscitated ancestral L1Pa5 [26]. The 14-heptad coiled coil was determined by Scorer 2, http://coiledcoils.chm.bris.ac.uk/Scorer/ [62] numbering the heptads 1–14 starting from the amino terminus and using the amino acid after the end of heptad 14 as the beginning of the carboxy-terminal half of ORF1p. (Note that heptads were numbered in the opposite order in references [40, 45]). The coiled coil alignments were refined by reference to the encoded peptide sequences, and CG dinucleotides were restored in the 50% consensus sequences at positions of the alignments that were populated by CG, and TG or CA (usually both). As the L1 families are of different ages, they will have undergone time-dependent decay of their CG dinucleotides [63]. To minimize the

confounding effect of this decay on coiled coil amino acid variation, we translated the amino acid positions encoded by CG-affected codons (*i.e.*, CGN, NCG, NNC•GNN) to the null amino acid character, "O". We then converted amino acids at the corresponding positions of the aligned coiled coil peptides to "O", (subsequently converted to a "-", *i.e.*, missing data, by downstream processing) prior to principal component and cluster analysis. We refer to such sequences as CG-null and add "_o_" to the name of 50% consensus sequences derived from these peptides. S5 Fig shows the 50% consensus peptides for each retrieved cluster (*e.g.*, 1.7_o, cluster 1 of L1Pa7, see Fig 3) which were used for phylogenetic analysis (see next section). These consensus peptides (where X indicates positions which did not reach the 50% threshold) are ordered according to their position on the phylogenetic tree (Fig 4). The corresponding consensus peptides with the null amino acid restored to its original value are referred to as CG-restored and have an "_rt" added to their name (*e.g.*, 1.7rt). These sequences are also shown in S5 Fig and were used to construct a sequence LOGO of the clusters [64] (http://weblogo.berkeley.edu). The LOGO plot in S6 Fig shows that more than one position in every heptad except number 14, was subject to variation. Alignments of ORF1p and ORF2p sequences and those of the coiled coil domains, and clusters thereof (see next section), CG-less translations of the aligned clusters and the C-terminal half of ORF1p are supplied in S1 Data. The Perl script for "CG-less" translation, cg_less_trans.4f.1.pl, its rationale, and a test input file have been deposited to GitHub (https://github.com/anthony-f/avf_perl). Routine sequence editing and display were also carried out using EMBOSS as implemented at <http://bioinfo.nhri.org.tw/gui/>.

## Determination of coiled coil sequence space

To determine the coiled coil sequence space of the L1Pa7 –L1Pa1 families we subjected alignments of their coiled coils to two analytical techniques that provide three- or two-dimensional graphical views of sequence relatedness. The first was principal component analysis (PCA) on binary vectors (one-hot) encoding the appearance of amino acids at each position in the sequence in a global alignment of all the coiled coils of all the families in an unbiased manner with no family differences taken into account. One hot encoding assigns to each amino acid a unique 20-bit vector. No specific properties of amino acids were taken into account, but implicit in the one-hot encoding is a set of constraints that enforce the fact that there is only one specific amino acid at every position in any sequence. In particular, PCA does not require a choice of distance or similarity measure between sequences beyond the vector space structure implicit in the one-hot encoding. The python script (furano_v5.py) for PCA using one hot coding and the input alignments have been deposited to GitHub (https://github.com/nihcompmed/furano).

We also carried out K-means based cluster analysis on the coiled coil and C-terminal half of each family using the Bios2mds R package [50]. This package uses metric multidimensional scaling (MMDS) to visualize in two dimensions the sequence differences between the aligned coiled coils for each family. Closely related sequences appear as clusters and this package provides various functions for additional analysis such as sequence retrieval from each cluster. We used RaXml [56] running on the NIH High Performance Cluster (HPC) system to carry out phylogenetic analyses on the 50% consensus sequences of these clusters and visualized the trees with Dendroscope 3 [65]. We also used the usearch V11 suite to probe the ORF1 coiled coils of L1Pa1, L1pa2, and their immediate ancestor, cluster 1 of L1pa3 (cL1.3) for emerging sub-clusters by using its cluster_fast command with an -id (identity) parameter set to 1.00[57]. This would recover coiled coil clusters that are minimally divergent from their consensus coiled coils.

## Supporting information

**S1 Fig. ORF1p variant expression in HeLa cells.** Fifty μg samples of extracts from HeLa cells expressing C-terminally FLAG-tagged ORF1p of the indicated constructs were subject to denaturing gel electrophoresis and Western blotting with anti-Flag and anti-tubulin antibodies as described in the Materials and Methods / **Western Blot Analysis**. The * indicates ORF1p constructs that were active for retrotransposition.
(PDF)

**S2 Fig. Retrotransposition assays of ORF1p variants.** These assays are in addition to those shown in Fig 1C carried out as described in Materials and Methods / **Retrotransposition assays**. To reduce the size of the file, stained cell foci are only shown for the retrotranspositions in panel A.
(PDF)

**S3 Fig. ORF1p variants.** The full version of Fig 1B.
(PDF)

**S4 Fig. C-term clusters.** Alignments of the C-terminal half (see Fig 1A) of L1Pa7 –L1Pa3 were analyzed using the Bios2mds R package [50] as described in the Materials and Methods.
(PDF)

**S5 Fig. Alignment cluster consensus sequences.** The peptides of the indicated cluster consensus without the CG-affected amino acids (CG-null), indicated by _o, vs. the consensus sequence with the amino acids encoded by CG-affected sites restored, rt.
(PDF)

**S6 Fig. LOGO plot.** LOGO plot of 50% consensus coiled coil peptide sequences, with CG positions restored, labeled rt in S5 Fig. Arrow heads indicate position of the ancestral amino acids that are negatively epistatic in the modern context.
(PDF)

**S7 Fig. L1Pa1_cc_align.** Alignment of L1Pa1 coiled coil peptide sequences vs the 50% consensus sequence of the CG-null L1Pa1 coiled coil.
(PDF)

**S8 Fig. L1Pa2_cc_align.** Alignment of L1Pa2 coiled coil peptide sequences vs the 50% consensus sequence of the CG-null L1Pa2 coiled coil.
(PDF)

**S9 Fig. L1Pa2_ccF134_align.** Alignment of L1Pa2 CG-null coiled coil peptide sequences that have an F at position 134 vs the 50% consensus sequence of L1Pa2.
(PDF)

**S10 Fig. L1Pa2_cc_nonF134_align.** Alignment of L1Pa2 CG-null coiled coil peptide sequences that lack F at position 134 vs the 50% consensus sequence of L1Pa2.
(PDF)

**S11 Fig. cL1.3_cc_align.** Alignment of the L1Pa3 cluster1 coiled coil peptide sequences vs the 50% consensus sequence of the CG-null cL1.3 coiled coil and the 50% consensus sequences of the L1Pa1 and L1pa2 coiled coils.
(PDF)

**S1 Data. Compendium of all of the ORF1 and ORF2 sequences and their provenance used in this paper, an explanatory README file, and the annotated sequence of the**

**retrotransposition vector, pRTC2L1.3.txt (zipped file).**
(ZIP)

## Acknowledgments

This work utilized the computational resources of the NIH HPC Biowulf cluster. (http://hpc.nih.gov).

## Author Contributions

**Conceptualization:** Anthony V. Furano, Charlie E. Jones, Pamela R. Cook.

**Formal analysis:** Anthony V. Furano, Vipul Periwal, Jean-Claude Walser.

**Funding acquisition:** Anthony V. Furano.

**Investigation:** Anthony V. Furano, Charlie E. Jones, Kathryn E. Callahan, Jean-Claude Walser, Pamela R. Cook.

**Methodology:** Anthony V. Furano, Charlie E. Jones, Pamela R. Cook.

**Project administration:** Anthony V. Furano, Pamela R. Cook.

**Resources:** Anthony V. Furano, Kathryn E. Callahan, Jean-Claude Walser.

**Software:** Anthony V. Furano, Vipul Periwal, Jean-Claude Walser.

**Supervision:** Anthony V. Furano, Charlie E. Jones.

**Validation:** Anthony V. Furano, Charlie E. Jones.

**Visualization:** Anthony V. Furano, Vipul Periwal.

**Writing – original draft:** Anthony V. Furano, Charlie E. Jones.

**Writing – review & editing:** Anthony V. Furano, Charlie E. Jones, Jean-Claude Walser, Pamela R. Cook.

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
