## [Decision Letter · Decision Letter 0]

20 May 2020

Dear Dr Furano,

Thank you very much for submitting your Research Article entitled 'CRYPTIC GENETIC VARIATION ENHANCES PRIMATE L1 RETROTRANSPOSON SURVIVAL BY ENLARGING THE FUNCTIONAL COILED COIL SEQUENCE SPACE OF ORF1p' to PLOS Genetics. Your manuscript was fully evaluated at the editorial level and by independent peer reviewers. The reviewers appreciated the attention to an important problem, but raised some substantial concerns about the current manuscript. Based on the reviews, we will not be able to accept this version of the manuscript, but we would be willing to review again a much-revised version. We cannot, of course, promise publication at that time.

In particular, Reviewer #1 expresses substantial criticism about the presentation and writing quality (notably the excessive use of jargon). These issues would require thorough revisions to the text and narrative/rationale for the experiments in order to make the paper more broadly accessible outside of the L1 community.      

If you decide to revise the manuscript for further consideration at PLOS Genetics, please aim to resubmit within the next 60 days, unless it will take extra time to address the concerns of the reviewers, in which case we would appreciate an expected resubmission date by email to plosgenetics@plos.org.

[LINK]

We are sorry that we cannot be more positive about your manuscript at this stage. Please do not hesitate to contact us if you have any concerns or questions.

Yours sincerely,

Cédric Feschotte

Associate Editor

PLOS Genetics

Bret Payseur

Section Editor: Evolution

PLOS Genetics

Reviewer's Responses to Questions

**Comments to the Authors:**

Reviewer #1: Furano and colleagues present an extension of previous work from their lab (Nabuan Naufer, et al 2016) surrounding the fascinating question of what drove the rapid evolution of L1 ORF1p in early primate evolution. In their previous work, the group observed loss of retrotransposition activity from a chimeric ORF1 comprised of segments from modern (L1HS/PA1) and ancient (PA5) L1s and nicely demonstrated that this loss of activity in the chimera derived from a comprised rate of ORF1p trimerization. In the present manuscript, the group presents a more thorough mutagenesis of ORF1 guided by the changes that occurred between the PA5 and PA1 consensuses. The authors create numerous chimeras and targeted mutants and measure the retrotransposition activity of each. Generally, the data support that there are mutations that have no measurable effect on in vitro retrotransposition, mutations that have moderate effect, mutations that inactivate, and mutations for which their effect of function differs based on the sequence of the rest of ORF1. The logic for why the authors chose the specific mutations they present and any deeper interpretations of interactions amongst mutations is lost in a dense and confusing narrative and presentation of the data. The subsequent analyses of the sequence space of L1 subfamilies was unclearly motivated and hard to follow, and the conclusions made from these data overreached. Overall, the manuscript’s density, jargon, and lack of narrative make it a tough fit for PLOS Genetics. With a clearer motivation of each experiment and a tempering of conclusions, this paper would be a reasonable addition to the literature in a more specialized journal.

Major Points:

1) The motivation for carrying out each experiment is unstated, and the results section is extremely abrupt in its opening. A clear presentation of the reason and rationale for each experiment would be very helpful.

2) The authors seem to confound the accepted use of several distinct terms as used in the protein evolution field. First, it has been widely shown in many types of systems that natural proteins tend to be tolerant of most mutations (robustness) but also capable of functional change upon mutation at specific positions which comprise a minority of the sequence (adaptability or evolvability). This robust state of most proteins may result in the accumulation of mutations which are neutral in the context of the present fitness function, but upon some change (environmental, sequence change elsewhere, etc) become advantageous – these now adaptive changes were cryptic. The relevance of the concept of cryptic variation to the manuscript is unclear, and the above terms are oft used inappropriately. For example, (line 134) “Genetic robustness would be permissive to the accumulation of cryptic genetic changes - termed “evolvability”…”. There are many definitions of evolvability, but evolvability and cryptic variation are not synonymous. The relevance of this concept should be much more clearly elaborated.

Further, it seems that (ln 225) “relative indifference of L1 retrotransposition activity to extensive ORF1p coiled coil amino acid substitutions” and the (ln 228) “expansion of the functional coiled coil sequence space” are the same. Protein robustness is the expansion of functional sequence space via “indifference” to mutation.

To clarify these statements, it would be useful to reference more of the extensive literature on protein robustness and evolvability and provide a deeper discussion of how these topics relate to the data presented.

3) The finding of context dependence of mutations (epistasis) within ORF1p is not surprising; it likely exists in every protein. It could be interesting to interpret the observed epistasis in the context of the CC structure to try to find mechanistic explanation of the observed effects. Indeed, the manuscript would benefit from a more thorough discussion of how these findings should be interpreted in the context of the ORF1p structure (Khazina, et al. NSMB 2011), and how the present manuscript differs from the mutagenesis data presented in that paper.

Minor Points:

1) The figure are generally very dense and hard for me to parse. I suggest minimizing the number of colors and sequences to highlight the most important points (eg, a 25 seq alignment (fig 1B) and >20 colors (fig 4))

2) The construct names are non-intuitive and make any comparison of sequence differences vs activity difference tough. (fig 1)

3) If the conclusion of the PCA is that ORF1 is robust, it seems that you don’t need PCA to observe and quantify variation in this sequence. (ln 155)

4) The observation that “the paired active and inactive variants share the same sequence space” does not demonstrate epistasis (ln 156); it is well-known that single mutations can completely inactivate with little dependence on the rest of the sequence (for example, a highly thermodynamically destabilizing mutation).

5) The title and introduction needs much less jargon; specifically, the title borrow complex jargon from several fields, making it hard to understand for specialists from many fields.

6) In the discussion, cis preference is mentioned to support why hybrid trimer formation would not be the driver of having coexisting versions of coil coiled domains within a LINE1 family. Cis preference is known for ORF2p binding to LINE1 mRNA, but this is not necessarily the case for ORF1p?

Reviewer #2: This work is from a group that in the past has published seminal work on the domain functions of L1 ORF1p and on the evolution of LINE-1 families in both human and non-human genomes. As noted by the authors, variations in the ORF1p coiled-coil domain have previously been interpreted to adaptive evolution by the L1. In this study, Furano and group, expand significantly on the earlier studies to explore in detail the evolutionary dynamics of the coiled-coil domain of ORF1p, relating this to retrotransposition function in an established cell culture reporter assay, and demonstrating a couple of genetic phenomena along the way.

Nevertheless, this study provides no insights into the mechanism effects of the various mutations and expansions of the coiled coil domain of ORF1p, beyond the cell culture assay -- and this assay is a final readout of cumulative mechanisms of retrotransposition and does not itself provide mechanistic insights. Possible experiments might explore effects on chaperone activity, trimerization, or subcellular localization using immunomicroscopy. The authors hint at such experiments in the discussion but do not attempt them. Furthermore, as cited, this group has also previously reported that extensive amino acid substitutions have occurred during evolution of the L1Pa7-L1Pa3 families.

The question is, therefore, is the present detailed study of the ORF1 coiled coil of sufficient interest and scope for the readers of PLoS Genetics. While perhaps somewhat debatable, I would say yes. Using multiple and mutually supporting computational approaches, the study teases out the evolutionary dynamics of an essential cis protein for retrotransposition of the LINE-1, which comprises at least 17 percent of the human genome. The authors clearly show epistasis (eg., the effect of I77L) and genetic robustness of the coiled coil domain. They demonstrate in considerable detail the varied evolutionary history of the CC domain of ORF1, distinct from that of the c-terminal domain of ORF1p and of ORF2p. Phylogenetic and PCA analyses also allowed the authors to identify distinct coiled-coil clusters and to reveal their persistence through L1 family evolution.

There are a number points that warrant further clarification/elaboration prior to publication:

On line 101 the previously described (ref. 26) "resuscitated" L1Pa5 ORF1p is noted. A sentence or two on how the resuscitation was originally engineered would help reader understanding.

On line 114 and in Fig. 1, it is stated that ORF1 proteins are expressed at about the same extent in HeLa cells. This is clearly not the case - there would appear to be up to a 3-fold difference, especially obvious in the case of L107F. Properly, band intensities of non-saturating Western blot exposures should be measured, with ORF1p level presented as a ration of tubulin expression. Accurate measurements of expression levels (and also transfection efficiencies) are important in assessing retrotransposition assay results.

The antibiotic-based retrotransposition assay used here has traditionally been run by first selecting cells on hygromycin for transfection, and then on G418 to select for retrotransposition. The readout is then number of retrotransposition events/number of transfected cells, which controls for variations in transfection efficiencies. This paper failed to select on hygromycin, therefore a discussion of this fact and any possible bias should be included.

Clarity of the data would be increased by reproducing the retrotransposition frequency data in an additional column for each of the constructs described in Fig. 1B.

Line 181: I failed to find Supporting Data/orf1_FL/l1pa5. Is reference being made to a portion of a different file?

Line 195: I was confused by reference to the "circled node" in Fig. 4. Do you mean the colored circles?

Line 262: the cis-preference model is somewhat misstated, and the sentence should better read: "... whereby retrotransposition-competent L1 proteins bind their encoding transcript". The early experiments on cis-preference indicated that cis-binding was a component of successful retrotransposition, not that L1 proteins could not bind in trans to non-retrotransposition intermediates.

Line 265. It is stated: "Furthermore, there is evidence that translation can be sequestered in subcellular compartments or translation factories (eg. 49)". The relevance of this specifically to the L1 is unclear.

Methods, line 309. A number of modifications were made to the retrotransposition vector pRTC2, including moving the reporter cassette and swapping in new promoters. Explanations of why these changes were made would be of interest to members of the field.

Reviewer #3: In the manuscript entitle “Cryptic genetic variation enhances primate L1 retrotransposon survival by enlarging the functional coiled coil sequence space of ORF1p” the authors present an original analysis of the human LINE-1 ORF1p coiled coil domain. They used bioinformatic tools to study the expansion of the sequence space of the coiled-coil functional domain.

This manuscript is very well written. The work presented here is of great interest in the field retrotransposon biology, it contributes to our understanding on the variability of the N terminal domain of ORF1p to maintain its activity. It would also contribute to a general understanding on coiled coil domain variability of any protein.

One remaining question could be addressed in more detail in the discussion. This concern line 250 to 255 page 9. Based on the results, the authors suggest that different coiled coil sequences amplified concurrently through several subfamilies. Then, how to explain the presence of several coiled coil domains shared in different subfamilies, since these subfamilies, based on ORF2 phylogeny, appeared sequentially in a unique linear history? It would imply that the coiled coil domain of ORF1p (or the entire ORF1p sequence) can be exchanged between L1 subfamilies. It could also suggest, less likely, convergent mutations of the coiled coil sequence in different subfamilies.

Minor comments

Line 113/114, replace S1 Fig 1 by S2 Fig 2 and vice versa

Line 254, Figs 4 and 5 instead of 5 and 6

In figure 2 and 3, for younger subfamilies L1Pa1 L1Pa2, what is the explanation for the dots that are isolated from the others of the same subfamily?

Line 181, Supporting Data/orf1_FL was not provided

**Have all data underlying the figures and results presented in the manuscript been provided?**

Reviewer #1: Yes

Reviewer #2: Yes

Reviewer #3: None

PLOS authors have the option to publish the peer review history of their article (what does this mean?). If published, this will include your full peer review and any attached files.

Reviewer #1: No

Reviewer #2: No

Reviewer #3: No

---

## [Editor Report · Decision Letter 1]

13 Jul 2020

Dear Dr Furano,

We are pleased to inform you that your manuscript entitled "CRYPTIC GENETIC VARIATION ENHANCES PRIMATE L1 RETROTRANSPOSON SURVIVAL BY ENLARGING THE FUNCTIONAL COILED COIL SEQUENCE SPACE OF ORF1p" has been editorially accepted for publication in PLOS Genetics. Congratulations!

Yours sincerely,

Cédric Feschotte

Associate Editor

PLOS Genetics

Bret Payseur

Section Editor: Evolution

PLOS Genetics

Comments from the reviewers (if applicable):

**Data Deposition**

http://datadryad.org/submit?journalID=pgenetics&manu=PGENETICS-D-20-00460R1

**Press Queries**

---

## [Editor Report · Acceptance letter]

5 Aug 2020

PGENETICS-D-20-00460R1 

CRYPTIC GENETIC VARIATION ENHANCES PRIMATE L1 RETROTRANSPOSON SURVIVAL BY ENLARGING THE FUNCTIONAL COILED COIL SEQUENCE SPACE OF ORF1p 

Dear Dr Furano, 

We are pleased to inform you that your manuscript entitled "CRYPTIC GENETIC VARIATION ENHANCES PRIMATE L1 RETROTRANSPOSON SURVIVAL BY ENLARGING THE FUNCTIONAL COILED COIL SEQUENCE SPACE OF ORF1p" has been formally accepted for publication in PLOS Genetics! Your manuscript is now with our production department and you will be notified of the publication date in due course.

With kind regards,

Matt Lyles

PLOS Genetics

On behalf of:
